# Clustered Regularly Interspaced Short Palindromic Repeats in *Xanthomonas citri*—Witnesses to a Global Expansion of a Bacterial Pathogen over Time

**DOI:** 10.3390/microorganisms10091715

**Published:** 2022-08-26

**Authors:** Ninon Bellanger, Alexis Dereeper, Ralf Koebnik

**Affiliations:** Plant Health Institute of Montpellier, University of Montpellier, Cirad, INRAE, Institut Agro, IRD, 34394 Montpellier, France

**Keywords:** *Xanthomonas citri*, citrus canker, CRISPR, ancient DNA, spoligotyping, epidemiology, evolution

## Abstract

*Xanthomonas citri* pv. *citri*, a Gram-negative bacterium, is the causal agent of citrus canker, a significant threat to citrus production. Understanding of global expansion of the pathogen and monitoring introduction into new regions are of interest for integrated disease management at the local and global level. Genetic diversity can be assessed using genomic approaches or information from partial gene sequences, satellite markers or clustered regularly interspaced short palindromic repeats (CRISPR). Here, we compared CRISPR loci from 355 strains of *X. citri* pv. *citri*, including a sample from ancient DNA, and generated the genealogy of the spoligotypes, i.e., the absence/presence patterns of CRISPR spacers. We identified 26 novel spoligotypes and constructed their likely evolutionary trajectory based on the whole-genome information. Moreover, we analyzed ~30 additional pathovars of *X. citri* and found that the oldest part of the CRISPR array was present in the ancestor of several pathovars of *X. citri*. This work presents a framework for further analyses of CRISPR loci and allows drawing conclusions about the global spread of the citrus canker pathogen, as exemplified by two introductions in West Africa.

## 1. Introduction

Xanthomonads are a large genus of Gram-negative, plant-associated bacteria that shows a high degree of host plant specificity. Pathogenic members of the genus cause diseases on over 400 host plants such as rice, citrus, cassava, pepper, wheat, banana, cabbage, tomato and bean. Many of these bacteria cause significant yield losses of economically important crops, such as cereals, solanaceous and brassicaceous plants. They cause a variety of symptoms, including necrosis, cankers, spots, and blight, and they affect different parts of the plant, including leaves, stems, and fruits. The genus currently comprises 31 validly described species [https://lpsn.dsmz.de/search?word=Xanthomonas, accessed on 12 April 2022] and is further divided into subspecies (subsp.; based on genetic criteria) or pathovars (pv.; based on phytopathological data). Global and local epidemiologic surveillance is applied to this pathogen because it represents a major threat for agricultural industries worldwide [1].

Collectively, strains of *Xanthomonas citri* can infect many different host plants, among which Citrus is one of the economically most important host plants. Citrus is a genus of flowering plants belonging to the family Rutaceae. It originated in Australia, New Caledonia, and Southeast Asia [2]. Citrus fruits provide an ample supply of vitamin C, folic acid, minerals, fiber, and various phytochemicals such as carotenoids, flavonoids, and limonoids, which have tremendous health benefit.

The production of citrus fruits, however, is threatened by bacterial canker disease, which is caused by *X. citri* pv. *citri*. Symptoms include leaf spotting, fruit rind blemishing, defoliation, shoot dieback, and fruit drop under favorable environmental conditions conducive to pathogen proliferation. This disease has a serious economic impact on citrus production worldwide. Citrus canker outbreaks occurred in the state of Florida in 1910, 1984, and 1995 [3].

Citrus canker is caused by three pathotypes for *X. citri* pv. *citri*, A, A*, and A^w^, which have been assigned based on host specificity and defense responses triggered on some of their host plants [4]. Pathotype A is the most widely spread pathotype, which causes severe disease on Citrus and related species of the Rutaceae family. Pathotype A* and A^w^ have a comparatively narrower host range, infecting key lime (*Citrus aurantifolia*) and alemow (*Citrus macrophylla*), and either not producing symptoms on grapefruit (A*) or eliciting a hypersensitive response on grapefruit (A^w^).

*X. citri* pv. *citri* is not the only pathovar known in *X. citri*. Other well-known pathovars are *glycines*, *mangiferaeindicae*, *malvacearum*, *phaseoli*, *vignicola*, members of which infect soybean, mango, cotton, bean and cowpea, respectively. Currently, ~30 pathovars of *X. citri* have been described, including bacteria that infect grapevine, pomegranate, ornamental, medicinal or woody plants [5,6].

Bacteria are exposed to infectious entities as well, such as bacteriophages or other hostile bacteria. Yet, they can persist well in a multitude of ecological systems, thereby sometimes relying on clustered regularly interspaced short palindromic repeats (CRISPRs). CRISPRs constitute defense mechanism against parasitic organisms. Since their first discovery in *Escherichia coli* in 1987, they were found within the genomes of ~40% of bacteria and ~90% of archaea [7]. Any particular bacterial species may contain more than one CRISPR locus whereas others do not have this defense mechanism and may rely on other resistance mechanisms.

Most CRISPR loci consist of a CRISPR array and set of CRISPR-associated (*cas*) genes. *cas* genes belong to large and polymorphic gene families, whose members encode proteins that carry nucleic acid-related domains such as nucleases, helicases, and nucleotide binding motifs. Six ‘core’ *cas* genes are known, *cas1* to *cas6*, among which the strongly conserved *cas1* gene can be considered a universal marker for CRISPR/*cas* systems [7]. CRISPR arrays typically consist of several quasi-identical noncontiguous, partially palindromic DNA sequences (direct repeats) that are separated by stretches of nonrepetitive sequences (spacers). The size of the direct repeats (DRs) and CRISPR spacers varies between 23 to 47 base pairs (bp) and 21 to 72 bp, respectively [7].

The CRISPR leader, which is usually colinear with the *cas* genes and found upstream of the first CRISPR repeat, acts as a promoter for the CRISPR array [8]. CRISPR repeats are usually colinear with the *cas* genes, while terminal repeats often degenerate at the trailer side [7,9]. When cells are exposed to a bacteriophage, phage-resistant variants can evolve that incorporate one or multiple new CRISPR repeat-spacer units at the leader end, which makes the progeny then immune against this phage [7].

Depending on the architecture of the CRISPP locus, i.e., complement of *cas* genes, operon structure and repeat sequences, an updated classification was introduced in 2020, which includes 2 classes (1 and 2), 6 types (I to VI) and 33 subtypes [10]. Collectively, species of *Xanthomonas* have been described to possess two distinct CRISPR subtypes, IC and IF (formerly known as Dvulg and Ypest subtypes [8]), where some subspecies contain one or the other subtype, few contain both subtypes, and many do not contain any CRISPR locus [11,12]. Since CRISPR arrays can be considered rapidly evolving loci, they have been employed to develop molecular typing schemes [7,9].

Here, we performed genome-wide CRISPR analyses for the species *X. citri*, predicted all CRISPR arrays, compared and categorized all spacers, updated the CRISPR-based genealogy of *X. citri* pv. *citri* [13] and examined the CRISPR locus of an ancient *X. citri* pv. *citri* strain that was found in an herbarium specimen [14]. Specifically, we addressed the following questions: Do all strains of *X. citri* pv. *citri* have a CRISPR locus? Are their CRISPR arrays restricted to 23 spacers, as observed before [13]? Can CRISPR arrays be reconstructed from ancient DNA samples? Is the previously proposed genealogy of spoligotypes correct and can this scheme be improved be adding more samples? Is there a geographic signal in the spoligotypes? Do other pathovars of *X. citri* contain CRISPR loci? And if so, do they share spacers with each other, and where do the spacers come from?

## 2. Materials and Methods

### 2.1. Genomic Resources

Genome sequences for strains of *X. citri* were retrieved from NCBI GenBank, using ‘xanthomonas citri[orgn]’ as a query. Since we were aware of many additional GenBank entries that should belong to the *X. citri* species but that have been deposited as *X. campestris* or *X. axonopodis*, we also used these two species as queries. These mis-annotations are due to the fact that for a long time, strains of *Xanthomonas* were generally considered to belong to different pathovars of polytypic species *X. campestris* [15]. Other strains were mis-classified as *X. axonopodis* as members of the so-called Rademaker group 9, which was formed based on DNA-DNA reassociation studies and was thought to reflect a distinct bacterial species, *X. axonopodis* [16,17]. Rademaker group 9 was further divided into six subgroups, 9.1 to 9.6, and it was speculated that they may form different species [18]. Ten years later, five of the six Rademaker 9 subgroups were taxonomically reclassified into four species of *Xanthomonas*, *X. axonopodis* (PG IV; subgroup 9.3), *X. citri* (PG I; subgroups 9.5 and 9.6), *X. euvesicatoria* (PG II; subgroup 9.2), and *X. phaseoli* (PG III; subgroup 9.4), whereas subgroup 9.1 still awaits taxonomic reevaluation [19,20].

All relevant retrieved genome sequences, i.e., those annotated as *X. citri* and those of *X. campestris* and *X. axonopodis* that were not clearly belonging to these species, were taxonomically re-evaluated by calculating genome-wide average nucleotide identities (ANI) using the enve-omics webserver at http://enve-omics.ce.gatech.edu/enveomics/, accessed on 11 April 2022 [21]. All sequences that were at least 95% identical to the genome of the *X. citri* pathotype strain LMG 9322 (acc. no. CCVY01000000) were considered to belong to the species *X. citri*. This analysis was complemented by reassigning taxonomic information using the Type (Strain) Genome Server at https://tygs.dsmz.de, accessed on 13 April 2022, which calculates digital DNA:DNA hybrization values [22].

### 2.2. Prediction of CRISPR Loci in Genome Assemblies

Genome FASTA files were retrieved from NCBI GenBank and analyzed with the CRISPRCasFinder Perl script, as downloaded from https://github.com/dcouvin/CRISPRCasFinder on 11 May 2021, release 4.2.20 [23]. Since the two known CRISPR subtypes in *Xanthomonas* consist of direct repeats (DRs) of 28 or 31 bp, we restricted the search to DR sizes of 27 to 32 bp, thus reducing the number of false positives. Spacer and DR sequences were parsed from JSON output files into an Excel file. To exclude false positives, we manually scrutinized all predicted CRISPR arrays for the presence of bona fide CRISPR DR sequences.

### 2.3. Prediction of CRISPR Loci in a Sequence Read Archive (SRA)

CRISPR-related sequences from a historical herbarium-derived dataset were identified by BLASTN at NCBI (https://blast.ncbi.nlm.nih.gov/, accessed on 29 November 2021) with the corresponding Sequence Read Archive (SRX9261163; 220,879,176 sequences) as search set. As query sequences, we used the 37 known spacer sequences from *X. citri* pv. *citri* [13] as well as the three described repeat sequences (DR1, 5′-GGCGCGCCCTCACGGGCGCGTGGATTGAAAC; DR2, 5′-TTCGCGCCCTCATGGGCGCGTGGATTGAAAC; DR, 5′-GTCGCGCCCTCACGGGCGCGTGGATTGAAAC). To define the boundaries of the CRISPR array, we also used the conserved flanking sequences (100 bp), a.k.a. leader and terminator sequences, as queries.

Parameters were kept at default, except that the maximum target sequences were increased to 5000 and the filter for low complexity regions was deactivated. Notably, we kept the automatic adjustment for short input sequences active.

All BLASTN hits had a size of 150 nucleotides, consisting of the actual herbarium-derived sequence, a bar code used for DNA sequencing, the Illumina adaptor, and a stretch of A and G nucleotides as space holders to add to the total size of 150 bp. We used an in-house Python script to trim the non-herbarium-related nucleotides, starting with the barcode, and created a library of clean CRISPR-related sequences for further analyses.

## 3. Results

### 3.1. Curated Database of X. citri Genome Sequences

First of all, we established a curated database of *X. citri* pv. *citri* strains, for which genomic information was available at NCBI GenBank. Using ‘Xanthomonas citri’ as a query, we found 243 genome entries at GenBank, representing sequences from 228 different strains. In addition, we found another 251 strains of *X. citri* pv. *citri*, all from a recent genomic study of pathotype-A strains [24], which were not reported as genome entries at GenBank.

Because of known problems with taxonomic assignments at GenBank due to several taxonomic revisions, we also examined genome sequences of the species *X. axonopodis* and *X. campestris*. All entries that were not clearly assigned to known pathovars of these two species and which could potentially belong to the species *X. citri* were analysed by genome-wide ANI calculations and digital DNA:DNA hybrization. For ‘Xanthomonas campestris’, we found 175 entries at GenBank, ten of which were found to represent strains of *X. citri* (pvs. *centellae*, *leeana*, *merremiae*, *thespesiae*, *trichodesmae*, *vitiscarnosae*, *vitistrifoliae*, *vitiswoodrowii*). Likewise, for ‘Xanthomonas axonopodis, we found 20 entries at GenBank, eight of which were found to belong to the species *X. citri* (pvs. *bauhiniae*, *cajani*, *clitoriae*, *eucalyptorum*, *khayae*, *martyniicola*, *melhusii*). In addition, we found two additional strains of *X. citri* at GenBank, *X. campestris* pv. *azadirachtae* strain LMG 543 and *X. campestris* pv. *durantae* strain LMG 696, which were not reported as genome entries at the GenBank, probably because of their status of an ‘anomalous assembly’. We also included two strains of *Pseudomonas cissicola* (CCUG 18839, LMG21719) in our dataset since this species has been assigned to the genus *Xanthomonas* [25,26] and clearly belongs to the species *X. citri* based on genome-wide ANI and digital DNA:DNA hybrization calculations. Notably, the Type (Strain) Genome Server reports ‘Pseudomonas cissicola’ for Rademaker subgroup 9.5 *X. citri* strains and ‘Xanthomonas fuscans’ for Rademaker subgroup 9.6 *X. citri* strains.

In total, our curated dataset contained 516 genome sequences, among them 366 originating from *X. citri* pv. *citri*, and the rest from another 30 pathovars (Table 1 and Appendix A). Six strains of *X. citri* pv. *citri* were found to be sequenced twice, typically first by Illumina technology, followed by single-molecule long-read sequencing (LH201, LH276, JJ207-7, LL074-4, LM180, LM199). The pathotype strain LMG 9322 had even been sequenced three times, albeit once at very low quality (1143 contigs). Strain NCPPB 3615 is listed twice at GenBank but careful analysis revealed that the entry with accession number CDHC010000000 likely resulted from a mistake during the submission process and may correspond to strain LE116-1, which was another time submitted under accession number CDHD010000000. We also assumed that strains 306 and A306 as well as strains Xac29-1 and Xcc29-1 represent equivalent strains. Excluding these duplicates, we were left with 355 genome sequences of *X. citri* pv. *citri*.

A total of 150 sequences corresponded to other pathovars of *X. citri*, among them 48 from the bean pathovar *fuscans* (including seven duplicates), 18 from the soybean pathovar *glycines* (including three duplicates and one triplicate), twelve from the cotton pathovar *malvacearum* and twelve from the pomegranate pathovar *punicae* (including two duplicates). For 22 pathovars, only one or two draft genome sequences were available.

### 3.2. Inventory of CRISPR Spacers from Genome Sequences of X. citri pv. citri

FASTA files of these all 516 *X. citri* genomes were retrieved from GenBank and used to predict CRISPR loci and their spacer and DR sequences, using the CRISPRCasFinder Perl script. This algorithm scans nucleotide sequences for repeated sequence motifs, which are separated by distinct unique sequences (i.e., spacers). However, some proteins consist of repeated motifs as well, as for example a conserved predicted 218-aa surface protein in *Xanthomonas* (23-bp repeats), a filamentous haemagglutinin or the ice-nucleation protein [28], and are notoriously predicted as false positives. In order to reduce prediction of such false positives, we restricted the search to repeats between 27 and 32 base pairs in size, which includes the DR sizes of both CRISPR subtypes that are known for *Xanthomonas* (subtype IC, DR = 31 bp; subtype IF, DR = 28 bp) [11,12,29,30].

From those results, an inventory in the form of an Excel table was generated containing the strain name, the nucleotide sequence of the CRISPR spacer, the upstream and downstream DR sequences, and a unique ID for each CRISPR spacer that includes the name of the strain plus the order number of the spacer within the genome sequence (Appendix A).

This inventory contains all the spacers for each strain, in total 6447 for 355 strains of *X. citri* pv. *citri*. Manual inspection revealed ten false positives for eight strains (LG98_001, LI070-01_019, LI070-01_020, LK142-04_020, LM057-04_020, LM057-15_020, LM088-25_019, LM095-04_020, LM095-04_021, LP029-13_023), as inferred from the upstream and downstream DR sequences (Table 2 and Appendix A). Without the false positives, the average number of spacers per strain is 18.1. Except for one strain from Pakistan, CFBP 2911, none of the strains contained a spacer beyond the 23 canonical spacers that were previously described [13].

All strains of *X. citri* pv. *citri* were found to have a CRISPR array, which was in most cases encoded on a single contig. However, in a few strains, the CRISPR array was split into two parts due to the insertion of an IS element, as reported previously [13]. Since IS elements are present in several copies per genome, such events resulted in splitting of the CRISPR array into two contigs in case of Illumina-sequenced genomes (LB302, LG97, LG115, NCPPB 3608, X2003-3218). In most pathotype A^w^ strains (AW13, AW14, AW15, AW16, Aw12879, LB302, LG115, NCPPB 3608, TX160042, TX160197, X2003-3218), the IS element was inserted in the DR between spacers Xcc_20 and Xcc_21, whereas in strain LG97, the IS element was inserted in spacer Xcc_18.

We also observed cases of tandem duplications of single spacer-repeat units. For example, spacer Xcc_02 was sometimes found two (D07 (52), JA159-1 (52), JQ613-01 (52), LH201 (52), LH276 (52), LI214-09 (52), LJ207-7 (31), LL074-4 (52), LL082-03 (08), LL095-08 (08), LL098-02 (40), LL111-06 (52), LL124-01 (52), LL132-01 (52), LL174-05 (08), LL186-5 (08), LM089-02 (43), LM089-20 (52), LM090-02 (52), LM095-04 (19), LM096-08 (19), LM121-01 (52), LN006-18 (43)), three (D02 (08)) or even four (LL096-08 (52)) times. Spacer Xcc_12 was triplicated in strain JK004-04 (04) (spoligotypes given in brackets).

### 3.3. Spoligotypes of X. citri pv. citri

Since we had realized that none of the *X. citri* pv. *citri* strains contained a truly novel spacer beyond the 37 known ones [13], we used this information to tabulate the presence/absence of spacers for all genome sequences and to determine their spoligotypes, i.e., the presence/absence patterns of spacers. During evolution, CRISPR arrays ‘grow’ from the terminal side to the leader-proximal side. The oldest spacer is therefore called Xcc_01 and the youngest one Xcc_37 [13]. Since spacers Xcc_24 to Xcc_37 were only found in one strain, these spacers were not informative when comparing strains and therefore not considered for the spoligotypes.

From the CRISPRCasFinder output, we imported the strain names, the coordinates of the CRISPR arrays (including the two outer DR sequences), and the CRISPR array sequences into an Excel file (Appendix A). In cases where the array was split into two parts due to an IS element insertion, we extracted the entire region including the IS element from the genome sequence. This procedure was necessary because the CRISPRCasFinder program fails at reporting spacers that are next to the IS element when they are not flanked by DRs at both sides. Likewise, when the array was encoded on two contigs, we included the sequence beyond the predicted array portions until the contig end, thus ‘sampling’ missing spacers that were not flanked by DRs at both sides.

The established Excel file compares the known spacer sequences to the CRISPR array sequences and reports a ‘1′ in case of presence and a ‘0′ in case of absence (Appendix A). This pattern is then transformed into a binary code, where each code is specific to a certain spoligotype. We then compared known spoligotypes to the observed binary codes, which either assigned a spoligotype to the strain, or did not report any known spoligotype. In such a case, a novel spoligotype was assigned to this pattern. This procedure was repeated until all strains were associated with a defined spoligotype. During this procedure we observed that some spacers may have experienced mutations that typically resulted in single-nucleotide polymorphisms (Table 3). Whenever we observed such a sequence variation, we considered these variants as homologs and included them in the list of known spacer sequences as equivalents of the prototype spacer sequences.

We also observed cases where the direct repeats (DR) had undergone changes in sequence (Table 4). Whereas most of them likely resulted from point mutations, we also observed a case in strain FDC 828 where spacer Xcc_01 was lost due to recombination between the first and the second repeat, thus forming a hybrid between the two terminal, degenerate DRs.

In total, we observed 48 different spoligotypes in the set of 366 genome sequences (Appendix A). Four spoligotypes that were known from a previous PCR analysis [13], Xcc07 (LG116), Xcc13 (LE065-1), Xcc20 (JK148-10), Xcc23 (LH001-3), were not found in our dataset. Together with them, we provide evidence for the existence of 52 spoligotypes in a global collection of *X. citri* pv. *citri* strains. The distribution of spoligotypes is not uniform and likely strongly affected by sampling biases. There are two dominant spoligotypes in our dataset: spoligotype 08 was found 120 times and spoligotype 52 was found 61 times, thus representing half of all analyzed strains. Some spoligotypes were found with intermediate frequency: spoligotype 02 (18×), spoligotype 14 (37×), spoligotype 19 (16×), and spoligotype 38 (10×), whereas 20 spoligotypes were only observed once and eight spoligotypes were observed twice.

### 3.4. Genealogy of CRISPR Loci in X. citri pv. citri

Following a procedure that we had used before [13], we aimed at providing evolutionary scenarios of the spoligotypes for each of the three pathotypes of *X. citri* pv. *citri*, A, A* and A^w^. In brief, we used genomic information to calculate a robust phylogenetic tree, based on genome-wide average nucleotide identities (ANI), and placed the various spoligotypes in the corresponding phylogenetic tree. This procedure allowed reconstructing the evolutionary history of the spoligotypes. We chose one representative strain for each spoligotype and the corresponding phylogenetic tree was then associated with the observed spoligotypes (Figure 1). This figure shows that similar spoligotypes cluster in the phylogenetic tree. Since variation in spoligotype is due to recombinational loss of spacer/repeat units, this information helps to build hypotheses about the genealogy of the spoligotypes, i.e., which spacer/repeat units were lost first, etc.

We also calculated an ANI-based phylogenetic tree for all 28 strains belonging to pathotype A* (Appendix A). In total, ten spoligotypes were observed for this pathotype, with some spoligotypes only observed once (i.e., spoligotype 11, strain CFBP 2911; spoligotype 24, strain NCPPB 3615; spoligotype 49, strain AS8), whereas others were observed up to five times (i.e., spoligotypes 10, 16 and 18). Notably, strains with the same (or similar) spoligotype were isolated in the same geographical area. For instance, strains with spoligotype 18 originate from Iran and strains with spoligotypes 16 and 17 originate from Southeast Asia (Cambodia, Singapore, Thailand) (Appendix A).

Surprisingly, strains from pathotype A^w^ belonged to only three spoligotypes (01, 02, 04), two of which were also observed for pathotype-A strains (02, 04) (Appendix A). Since both pathotypes are postulated to have diverged early in diversification of the *X. çitri* pv. *citri* pathovar, we wondered if these shared spoligotypes may have evolved independently of each other. We therefor calculated the ANI-based phylogenetic tree for all pathotype-A^w^ strains and also for all the other strains that share their spoligotype with them (Appendix A). This analysis demonstrates that strains of the two pathotypes cluster together. Moreover, within each pathotype, strains with the same spoligotype cluster as well. As before, there is a clear geographic signal in the data, with all pathotype-A strains from the Seychelles sharing spoligotype 02 and those from the Maldives sharing spoligotype 04. Likewise, we observed three lineages for the pathotype-A^w^ strains, with each lineage corresponding to another spoligotype. Notably, each lineage includes strains from the US, an ancestor of which may have been introduced from another continent.

Based on the genomic information, we developed parsimonious models for the evolutionary scenarios of spoligotypes for all three pathotypes, thereby minimizing the number of postulated recombinational events, which can be deletion of one or several spacer/repeat units at once. Since pathotype A^w^ has only three spoligotypes, the most parsimonious scenario starts with spoligotype 01 (containing all 23 spacers). Deletion of spacer Xcc_14 then yielded spoligotype 02, and subsequent deletion of spacer Xcc_11 may have resulted in spoligotype 04. Close linkage of spoligotypes 01 and 02 is also supported by an IS element, that had jumped into the same position between spacers Xcc_20 and Xcc_21, resulting in a target site duplication of six base pairs [13]. However, since strains with spoligotype 04 do not contain the IS element at this position, yet share absence of spacer Xcc_14, we assume that deletion of spacer Xcc_14 in spoligotypes 02* and 04 were independent events. If so, we cannot conclude which of the two spacers in spoligotype 04 was lost first, Xcc_11 or Xcc_14.

Strains of pathotype A* fall into two major clades (Figure 2). The first clade contains strains from Cambodia, Singapore and Thailand. Here, spoligotype 17 originated from spoligotype 16 by deletion of spacer Xcc_05. The second major clade can be further divided in three subclades. Subclade 2A comprises strains from India, Oman, Pakistan and Saudi Arabia with spoligotypes 10, 11, and 24, where spoligotype 10 constitutes the predecessor of the other two spoligotypes. Subclade 2B includes strains from Iran and Saudi Arabia, where spoligotype 09 (Saudi Arabia) gave rise to spoligotype 49 (Saudi Arabia) by deletion of spacers Xcc_18/Xcc_19, which then evolved to spoligotype 18 (Iran) by deletion of spacers Xcc_03/Xcc_04. Subclade 2C contains strains from Ethiopia (spoligotype 12) and Iran (spoligotype 45; deletion of spacers Xcc_19 to Xcc_23). Based on these data, one may speculate that the strains in Iran resulted from two independent introductions.

The situation with pathotype A, for which we have the largest dataset, is much more complex (Figure 3). Yet, we succeeded to update the scheme that had been published before [13]. The previous scheme contained only four layers of recombinational events, with the fourth layer consisting of spoligotypes 14, 19, 20, 21, and 26 (green circles in Figure 3). Based on the genomic data, we can now add another eight spoligotypes to layer 4 and form two additional layers (eleven observed spoligotypes in layer 5 and five spoligotypes in layer 6). Compared to the scheme from 2019, where we had only 16 spoligotypes identified among 56 analysed strains (25 pathotype-A strains), we now present a robust genealogy for 39 spoligotypes.

### 3.5. Characterization of the CRISPR Locus from an Ancient X. citri pv. citri

Recently, the first genome of a bacterial plant pathogen was sequenced from a herbarium specimen [14]. DNA that was extracted from a canker-symptomatic citrus leaf sample, originating from Mauritius Island in the Indian Ocean and dating back to 1937, was used to generate the first historical genome of a bacterial crop pathogen from such a specimen. When we analysed this genome sequence with CRISPRCasFinder, it was reported to belong to spoligotype 14 (deletion of spacers Xcc_14, Xcc_08, Xcc_10/Xcc_11, Xcc_03). This spoligotype is identical to the one of strain 306, which served as a reference genome in the mapping-based assembly of the herbarium-derived genome sequence. It was therefore possible that the herbarium specimen may contain spacer that were not present in the reference genome and thus belong to another spoligotype.

We wondered whether the sequence reads can be used to reconstruct the CRISPR array of that historical sample. In order to identify candidate reads, we used the 37 described spacer sequences as queries at NCBI GenBank [13]. In addition, we used the three DR sequence variants and the two flanking regions of the CRISPR array as queries. With these queries, we detected only for 19 of the described spacer sequences a homolog in the herbarium dataset. None of the 14 more recently acquired spacers, which have only been found in one strain from Pakistan (spoligotype 11), were found. Among the 23 widely conserved spacers, we missed finding a homolog for spacers Xcc_08, Xcc_10, Xcc_11, Xcc_14. This pattern thus corresponded to spoligotype 08, which had been found before in pathotype-A strains from China, Reunion Island, and USA [13].

We also observed reads which contained subsequences of two neighboring spacers, thus confirming the order of most spacers, the deletion of spacers Xcc_08 and Xcc_10/Xcc_11, and the absence of any additional, hitherto not observed spacer (Appendix A). However, in this metagenomic sample, we also found reads that contained DR sequences, which were flanked by candidate spacer sequences likely originating from other bacteria. Indeed, only 1.2% of the reads were mapped on the *X. citri* pv. *citri* reference genome, whereas at least 5.2% of the reads belonged to other bacteria [14]. This number may even be much larger because 60.1% of the reads were not assigned to any taxon and may well contain many more bacterial sequences, including pieces of CRISPR arrays.

### 3.6. Analysis of CRISPR Loci in Other Pathovars of X. citri

The species *X. citri* contains more pathovars than the one infecting Citrus, among them pathovars that were reported to belong to the species *X. axonopodis* or *X. campestris* [6]. We wondered whether the presence of the CRISPR locus is restricted to only one pathovar or if we could identify CRISPR loci in other pathovars. We used the CRISPRCasFinder tool to scrutinize all genome sequences that belong to the species *X. citri* (Table 1). In addition to the pathovar *citri*, we also detected CRISPR loci in the pathovars *durantae*, *cajani*, *clitoriae*, *melhusi*, *punicae*, and *khayae* (Figure 4). Four of these pathovars, pvs. *citri*, *durantae*, *cajani*, and *clitoriae*, are phylogenetically close to each other (Figure 4). Like strains of the pathvar *citri*, *X. citri* pv. *durantae* strain LMG 696 from India contains a subset of the 23 widely conserved spacers. With spacers Xcc_03, Xcc_06, Xcc_09/Xcc_10, and Xcc_19 missing, it matches to spoligotype 10, which was also found in five pathotype-A* strains of *X. citri* pv. *citri*: DAR73910 (international intercept from India at Sydney Airport), FDC 1682 (Oman), JF90-2 (Oman), JM35-2 (Saudi Arabia), NCPPB 3607 (India).

Strain LMG 558 (*X. citri* pv. *cajani*) contains 26 spacers, twelve of which are shared with *X. citri* pv. *citri*, whereas the other spacers have not been identified before. Notably, four of the novel spacers are shared with strain LMG 9045 (*X. citri* pv. *clitoriae*). In addition, this strain contains six of the 23 canonical spacers of *X. citri* pv. *citri*. Strains of the other three pathovars have also a few spacers in common with *X. citri* pv. *citri*, albeit at a smaller number, and they share spacers with each other and with pv. *clitoriae*. Our dataset includes ten strains of *X. citri* pv. *punicae*, for which we describe three different spoligotypes.

In total, the set of strains beyond pathovar *citri* contains 103 hitherto undescribed spacers; 62% of them do not have significant hits at NCBI GenBank’s non-redundant database. Among those with homologs, 30% matched to bacteriophage sequences and 75% matched to sequences in the genomes of xanthomonads. Since 35 of the novel spacers are at positions between one of the 23 canonical *X. citri* pv. *citri* spacers and the terminator, they must have been there before the divergence of the different pathovars of *X. citri*. Hence, the common ancestor of these pathovars contained a CRISPR array significantly larger than that of the founder of the pathovar *citri* lineage.

## 4. Discussion

### 4.1. Genomics-Informed Analysis of X. citri pv. citri Doubles the Number of Known Spoligotypes and Allows Reconstructing Their Probable Evolutionary Trajectory/Provides Information about Lineage Descendance of CRISPR Loci

In this study, we have made use of the wealth of genomic resources that are available for xanthomonads. We analyzed all publicly available genome sequences of strains of *X. citri*, including several pathovars that were taxonomically misidentified until recently [6]. We observed that all strains of *X. citri* pv. *citri* have a CRISPR locus and their CRISPR arrays were found to be restricted to 23 spacers, as observed before [13]. This work allowed us to update the genealogy of CRISPR spoligotypes for the pathovar *citri*. In comparison to the most recent study with 25 spoligotypes, we describe 26 novel spoligotypes and assign all of them a parsimonious position in the framework for the genealogy of the citrus canker pathogen.

In addition to the spoligotypes, we also compiled other variations at the CRISPR loci of 355 strains of *X. citri* pv. *citri*, such as presence of IS elements, SNPs in spacers (Table 3) and repeats (Table 4), which provide further information about lineage descendance. As a note of caution, however, we need to emphasize that some of these sequence variations may have resulted from sequencing artifacts and need critical review and/or independent confirmation before making use of them.

### 4.2. The Spoligotype Genealogy Framework Contains Information on the Global Spread of the Citrus Canker Pathogen, as Exemplified by the Two Introductions in West Africa

In general, we observed geographical signals in the dataset. Strains that originate from the same country or one of its neighboring countries often share the same or a similar spoligotype. However, we also found trans-continental commonalities. For instance, strains from West Africa contain either spoligotype 14 (Mali) or 15 (Mali, Senegal). These two spoligotypes are quite distant to each other, suggesting at least two independent introductions of the pathogen in this area. Spoligotype 14 has been observed in 10% of the genome sequences, whereas spoligotype 15 was only found in four strains (LD7-1 and LE116-1 from Mali, LH37-1 from Senegal, NCPPB 3562 from India). These data are compatible with a previous study which used micro- and minisatellites to assess the geographic spread of the citrus canker pathogen [33]. It was suggested that two groups of strains, belonging to two DAPC (discriminant analysis of principal components) clusters, were introduced in Mali. DAPC-1 strains originated from very diverse geographic origins, which made it impossible to decipher the origin of West African DAPC-1 strains. Yet, Brazil was identified as the country that shared the largest number of haplotypes with strains from West Africa [33]. Interestingly, the Malian spoligotype 15-strain LC80 belongs to DAPC cluster 1, along with six strains from Brazil and eleven strains from Argentina that contain spoligotype 15 as well (Appendix A). DAPC-2 strains from Mali shared a haplotype with strains from Pakistan and it was suggested that DAPC-2 strains from Mali might have originated from the Indian subcontinent [33]. Again, the two Malian spoligotype-14 strains LD7-1 and LE116-1 belong to DAPC cluster 2, as does strain NCPPB 3562 from India, thus supporting the hypothesis of an introduction from the Indian subcontinent. In conclusion, our spoligotyping data are consistent with previous work using mini- and microsatellite analyses.

### 4.3. Metagenomic Data from a Herbarium Sample Enable Reconstruction of the Spoligotype of an Ancient Citrus Canker Pathogen and Its Positioning in the CRISPR-Based Genealogy

In this study, we succeeded in reconstructing the spoligotype of bacteria whose DNA was preserved in a herbarium specimen from 1937 originating from Mauritius [14]. BLAST analyses assigned this material to spoligotype 08, thus correcting the spoligotype which was assigned based on a reference-genome based assembly. Spoligotype 08 is the most dominant spoligotype in our set of 355 strains, which originate from Brazil, China, Comoros, Hong Kong, Mayotte, New Zealand, Reunion Island, Taiwan, USA, and Vietnam (Appendix A). It will be interesting to decipher the spoligotypes of additional, perhaps even older samples from similar sources.

### 4.4. The X. citri pv. citri CRISPR Locus Is Conserved in Some Additional Pathovars of X. citri and Different Pathovars Tend to Share Their Oldest Spacers

Phylogenomic analyses revealed that several *X. citri* pathovars beyond *citri* contain CRISPR loci. Interestingly, different pathovars share common spacers, indicating that their common ancestor harbored a CRISPR array that was significantly larger than those which are nowadays found in the pathovar *citri*. It appears that the whole locus got lost in several other pathovars, suggesting that the CRISPR locus is not required for inhabiting different plants and environments. We did not observe any linkage to certain modes of plant colonization or host plants. It appears that the locus got lost several times and was perhaps functionally substituted for by other defense systems against bacteriophages, such as restriction/modification systems.

One of the non-*citri* pathovar strains, i.e., LMG 696 (pv. *durantae*), did not contain any spacer that was not found in the pathovar *citri*, and therefore qualified as spoligotype 10. This strain was isolated from *Duranta repens* (family: Verbenaceae) and was considered a “clonal variant” of pathovar *citri* strains [34]. Moreover, this strain clustered in a maximum likelihood phylogenetic tree based on concatenation of 1785 single-copy genes with pathotype-A* strains FDC 1682, JF90-2, JM35-2, and NCPPB 3607, all belonging to spoligotype 10, and with pathotype-A* strain CFBP 2911, which belongs to the derived spoligotype 11 (Figure 2) [35]. These data are supported by our ANI-based phylogenetic tree, which clusters strain LMG 696 together with the pathovar *citri* pathotype strain LMG 9322 (Figure 4). It is not known whether strain LMG 696 can also infect plants of the Rutaceae family, as strains of pathovar *citri* do, and if it shares a similar host range as pathotype-A* strains [36]. Pathotyping on different candidate host plants and resequencing of strain LMG 696 using long-read technology could help clarify whether plasmids contribute to the different host ranges, as suggested previously [35].

### 4.5. Some Spacers Observed in X. citri Pathovars beyond the Citri Pathovar Originated from Bacteriophages

In contrast to the 23 canonical spacers of *X. citri* pv. *citri*, none of which was found to be homolog to contemporary bacteriophages at NCBI GenBank [13], we identified twelve spacers among the non-*citri* pathovar strains that are homologous to sequences from bacteriophages, mainly from *Xanthomonas*: Phage Cf (AJ011389.1), Cf1c (M57538.1), Cf1t (U08370.1), FoX2 (NC_055836.1), FoX3 (NC_055837.1), FoX4 (NC_055839.1), KPhi1 (NC_054460.1), M29 (MZ345003.1), MYK3 (OK275494.1), phiLF (MH206184.1), phiXaf18 (NC_054461.1), phiXv2 (MH206183.1), XacF1 (AB910602.), XAJ2 (KU197014.1), Xf109 (NC_043028.1), Xf409 (NC_055055.1). In addition, we observed hits with the genomes of the two *Xylella* bacteriophages Salvo (NC_042345.1) and Sano (NC_042344.1), with the *Stenotrophomonas* phage phiSMA6 (NC_043029.1), and with the Siphoviridae sp. isolate ctZLc1 (BK020633.1). We also observed that pathovars *clitoriae* and *melhusi* contained 23 and 33 spacers, respectively, that may have been acquired after divergence from the pathovar *citri*, as indicated by their leader-proximal localization with respect to the spacers that are shared with *X. citri* pv. *citri*. While we assume that the CRISPR locus in *X. citri* pv. *citri* is not able to acquire new spacers (perhaps except for strain CFBP 2911, which contains 14 extra spacers, some of them with homology to bacteriophages), we do not know if the locus in some of the other pathovars can acquire new spacers and provide resistance to contemporary bacteriophages. Analyses of a larger number of strains from these pathovars will help answering this question.

## 5. Conclusions

CRISPR arrays provide a novel approach to elucidate the evolutionary trajectory of bacteria, which have hitherto rarely been exploited for bacterial plant pathogens. Using *X. citri* as a model, we have demonstrated its usefulness to understand the evolution and global spread of the citrus canker pathogen. Moreover, conservation of CRISPR spacers in other pathovars of *X. citri* suggests that their common ancestor had a much larger CRISPR array than nowadays observed in strains of *X. citri*. It will be interesting to extend this study to additional species of *Xanthomonas* and perhaps even beyond them, once appropriate bioinformatic tools have been developed for such a large-scale analysis.

## Figures and Tables

**Figure 1 microorganisms-10-01715-f001:**
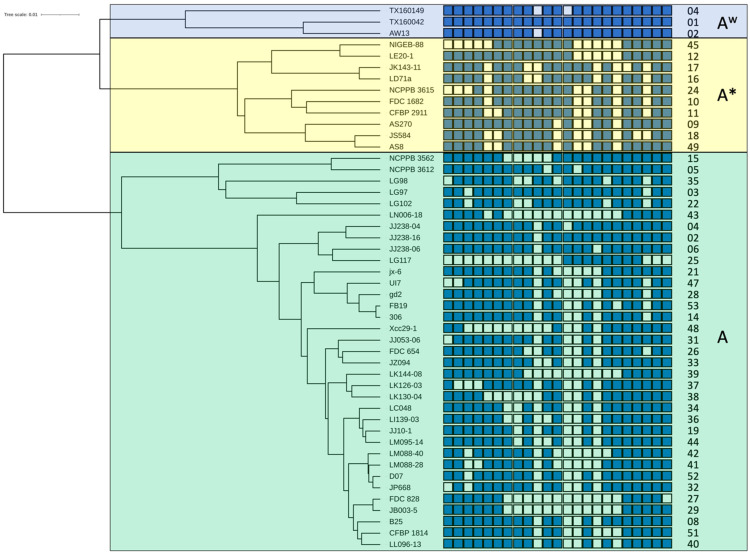
Phylogenetic tree of representative *X. citri* pv. *citri* strains and their associated spoligotypes. For each pathotype. i.e., A, A* and A^w^, one representative strain per spoligotype was included in the dataset to calculate genome-wide pairwise average nucleotide identities and perform their phylogenetic analysis on the enve-omics platform [21]. Complete (maximum) linkage clustering was used to build the phylogenetic tree [31]. The interactive Tree Of Life (iTOL) suite was used for better visualisation of the tree [32]. CRISPR arrays are represented by a series of 23 blue and open boxes, oriented with the leader-proximal spacers on the left side. Identical spacers within the same block are vertically aligned. Detected CRISPR spacers are represented by blue boxes. Open boxes indicate the absence of the corresponding spacer. Names of representative strains are shown between the tree and the CRISPR arrays. Spoligotype numbers are given on the right side. Transparent blue, yellow and green boxes frame pathotype-A^w^, pathotype-A*, and pathotype-A strains, respectively.

**Figure 2 microorganisms-10-01715-f002:**
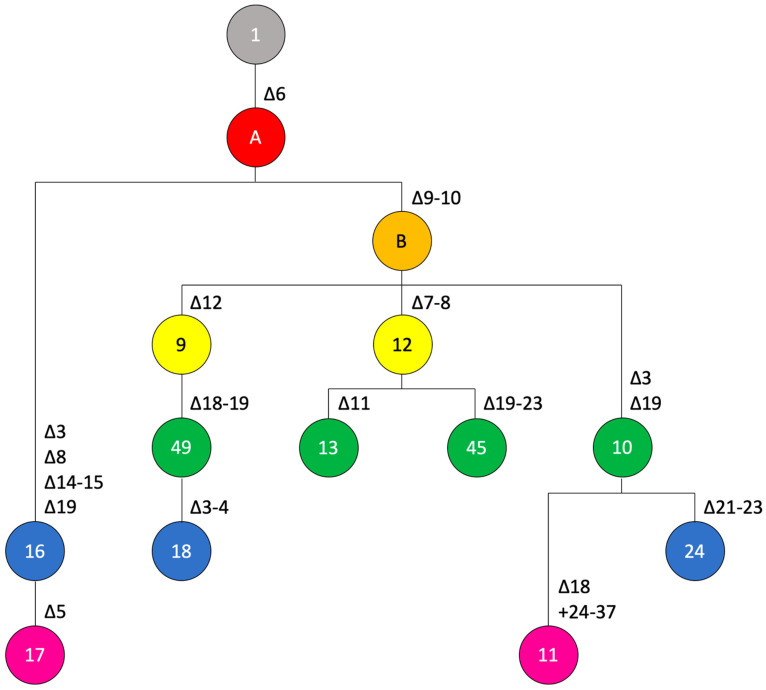
Genealogy of spoligotypes from pathotype-A* strains. Postulated mutational events leading to the observed spoligotypes are indicated, starting from the ancestral spoligotype with all 23 spacers shown in grey on the top, with the colors indicating the number of events (from one to six events, colored in salmon, orange, yellow, green, blue and pink, respectively). Numbers of spoligotypes are indicated in the circles. Characters, i.e., A and B, indicate postulated intermediate spoligotypes that were not observed among the analyzed pathotype-A* strains. Spoligotype 13 was not observed in any of the analyzed genome sequences but had been found in strain LE065-1 previously [13].

**Figure 3 microorganisms-10-01715-f003:**
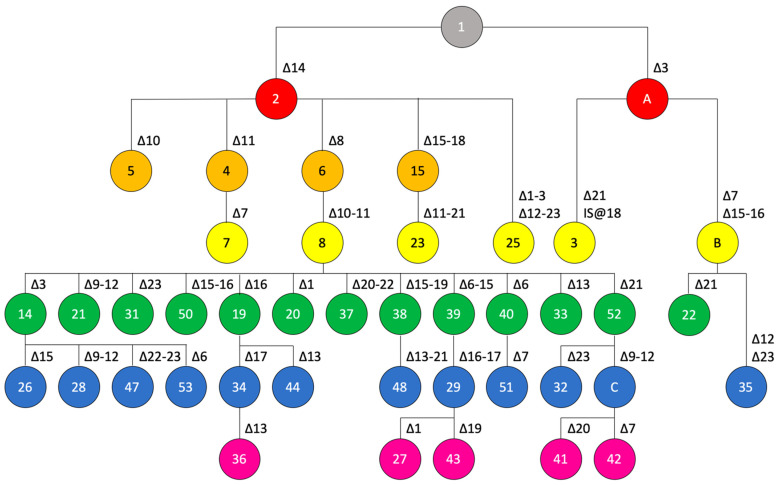
Genealogy of spoligotypes from pathotype-A strains. Postulated mutational events leading to the observed spoligotypes are indicated, starting from the ancestral spoligotype with all 23 spacers shown in grey on the top, with the colors indicating the number of events (from one to six events, colored in salmon, orange, yellow, green, blue, pink, respectively). Numbers of spoligotypes are indicated in the circles. Characters, i.e., A and B, indicate postulated intermediate spoligotypes that were not observed among the analyzed pathotype-A strains. Spoligotypes 7, 20, and 23 were not observed in any of the analyzed genome sequences but had been found before in strains LG116, JK148-10, LH001-3, respectively [13].

**Figure 4 microorganisms-10-01715-f004:**
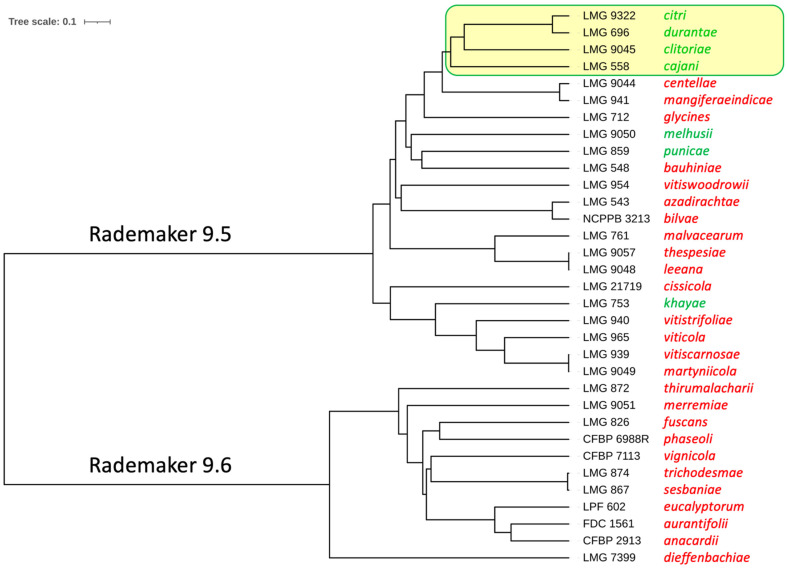
Phylogenetic tree of representative strains from 33 *X. citri* pathovars. Genome-wide pairwise average nucleotide identities were calculated, and their phylogenetic analysis was performed on the enve-omics platform [21]. Complete (maximum) linkage clustering was used to build the phylogenetic tree [31]. The interactive Tree Of Life (iTOL) suite was used for better visualisation of the tree [32]. 22 pathovars cluster within the Rademaker 9.5 group and 11 pathovars cluster within the Rademaker 9.6 group [18]. Strain names are shown to the left of the tree, followed by the pathovar name. Pathovars with CRISPR arrays are shown in green and those without CRISPR arrays in red.

**Table 1 microorganisms-10-01715-t001:** Overview of genomic resources used in this study and presence of CRISPR Cas systems.

Correct Taxonomic Status	GenBank Annotation	Subgroup ^a^	Complete Genome Sequences	Draft Genome Sequences	CRISPR Subtype ^b^
*X. citri* pv. *anacardii*	*X. citri* pv. *anacardii*	9.6	1	3	w/o
*X. citri* pv. *aurantifolii*	*X. citri* pv. *aurantifolii*	9.6	4	6	w/o
*X. citri* pv. *azadirachtae*	*X. campestris* pv. *azadirachtae*	9.5	0	1	w/o
*X. citri* pv. *bauhiniae*	*X. axonopodis* pv. *bauhiniae*	9.5	0	1	w/o
*X. citri* pv. *bilvae*	*X. citri* pv. *bilvae*	9.5	0	1	w/o
*X. citri* pv. *cajani*	*X. axonopodis* pv. *cajani*	9.5	0	1	IC
*X. citri* pv. *centellae*	*X. campestris* pv. *centellae*	9.5	0	1	w/o
*X. citri* pv. *citri*	*X. axonopodis*	9.5	1	0	IC
	*X. citri*	9.5	1	0	
	*X. citri* pv. *citri*	9.5	45	68	
	*X. citri* pv. *citri* ^c^	9.5	0	251	
*X. citri* pv. *clitoriae*	*X. axonopodis* pv. *clitoriae*	9.5	0	1	IC
*X. citri* pv. *dieffenbachiae*		9.6			w/o
*X. citri* pv. *durantae*	*X. campestris* pv. *durantae*	9.5	0	1	IC
*X. citri* pv. *eucalyptorum*	*X. axonopodis* pv. *eucalyptorum*	9.6	0	1	w/o
*X. citri* pv. *fuscans*	*X. citri* pv. *fuscans*	9.6	10	25	w/o
	*X. citri* pv. *phaseoli* var. *fuscans*	9.6	13	0	
*X. citri* pv. *glycines*	*X. citri* pv. *glycines*	9.5	12	6	w/o
*X. citri* pv. *khayae*	*X. axonopodis* pv. *khayae*	9.5	0	1	IC
*X. citri* pv. *leeana*	*X. campestris* pv. *leeana*	9.5	0	1	w/o
*X. citri* pv. *malvacearum*	*X. citri* pv. *malvacearum*	9.5	8	4	w/o
*X. citri* pv. *mangiferaeindicae*	*X. citri* pv. *mangiferaeindicae*	9.5	3	4	w/o
*X. citri* pv. *martyniicola*	*X. axonopodis* pv. *martyniicola*	9.5	0	1	w/o
*X. citri* pv. *melhusii*	*X. axonopodis* pv. *melhusii*	9.5	0	1	IC
*X. citri* pv. *merremiae*	*X. campestris* pv. *merremiae*	9.6	0	2	w/o
*X. citri* pv. *punicae*	*X. citri* pv. *punicae*	9.5	10	2	IC
*X. citri* pv. *sesbaniae*	*X. citri* pv. *sesbaniae*	9.6	0	1	w/o
*X. citri* pv. *thespesiae*	*X. campestris* pv. *thespesiae*	9.5	0	1	w/o
*X. citri* pv. *thirumalacharii*	*X. citri* pv. *thirumalacharii*	9.6	0	1	w/o
*X. citri* pv. *trichodesmae*	*X. campestris* pv. *trichodesmae*	9.6	0	2	w/o
*X. citri* pv. *vignicola*	*X. citri* pv. *vignicola*	9.6	3	0	w/o
*X. citri* pv. *viticola*	*X. citri* pv. *viticola*	9.5	0	2	w/o
*X. citri* pv. *vitiscarnosae*	*X. campestris* pv. *vitiscarnosae*	9.5	0	1	w/o
*X. citri* pv. *vitistrifoliae*	*X. campestris* pv. *vitistrifoliae*	9.5	0	1	w/o
*X. citri* pv. *vitiswoodrowii*	*X. campestris* pv. *vitiswoodrowii*	9.5	0	1	w/o
*X. citri* [pv. *cissicola*?]	*Pseudomonas cissicola*	9.5	0	2	w/o
*X. citri* not pv. *citri*	*X. citri*	9.5	0	5	w/o
*X. citri* not pv. *citri*	*X. citri*	9.6	4	1	w/o
Total			115	401	

^a^, Rademaker subgroup [18]; ^b^, Predicted CRISPR Cas subtype, w/o = without; ^c^, 251 strains taken from Richard et al. [27].

**Table 2 microorganisms-10-01715-t002:** Predicted false positive CRISPR arrays in *X. citri* pv. *citri*.

Strain	Spacer ID	DR Size (bp)	Spacer Size (bp)	Homolog ^a^
Canonical sequence		31	34–37	CRISPR array
LG98	Xanci2264	31	28	
LI070-01	Xanci2535 Xanci2536	27	28–29	Hypothetical protein in *Staphylococcus aureus* [8 × 10^−23^]
LK142-04	Xanci3512	27	43	Hypothetical protein in *Escherichia coli* [1 × 10^−13^]
LM057-04	Xanci4667	29	60	
LM057-15	Xanci4706	28	60	
LM088-25	Xanci4801	27	29	Hypothetical protein in *Staphylococcus aureus* [2 × 10^−11^]
LM095-04	Xanci4987 Xanci4988	29	19–22	
LP029-13	Xanci5825	28	62	

Loci that correspond to each other are highlighted by the same color. ^a^, Homologs were search at GenBank using TBLASTN, e values are given in square brackets.

**Table 3 microorganisms-10-01715-t003:** Variant spacer sequences in *X. citri* pv. *citri*.

Spacer	Variant Sequence	Canonical Sequence	Strains
Xcc_03a	AAGAAGACCAGTCTGCGGCGTCGCGGCATCCTTGGG	AAGAAGACCAGTCTGCGGCGTCGCGGCATCCTGGGG	JJ009-1
Xcc_03b	AAGAAGACCAGTCTGCGGCGTCGCGGCATCTTGGGG	AAGAAGACCAGTCTGCGGCGTCGCGGCATCCTGGGG	LL098-02
Xcc_03c	AAGAAGACCAGTCTGCGGCGTCGCGGCATCCTGGGGG	AAGAAGACCAGTCTGCGGCGTCGCGGCATCCTGGGG	LK136-05
Xcc_13a	GCCATCATGCTTTGAATGCGCCTACCCACGGCGAA	GCCATCATGCTTTGAATGCGCTTACCCACGGCGAA	UI6, UI7
Xcc_18a	GTGCCACCGACAGCGACGCACGTGGACCTGCATGTT	GTGCCACCGACAGCGACGCACGTGGACCTGCAGATC	LG97
Xcc_19a	TCGAGCGCATCGATGACGGTCACCCATCCCC_AATG	TCGAGCGCATCGATGACGGTCACCCATCCCCCAATG	LK169-03
Xcc_19b	GTGCCACCGATGACGGTCACCCATCCCCCAAT_	TCGAGCGCATCGATGACGGTCACCCATCCCCCAATG	jx4

Variant nucleotides in comparison to the canonical spacer sequence are double underlined.

**Table 4 microorganisms-10-01715-t004:** Variant DR sequences in *X. citri* pv. *citri*.

Strains	Sequence	Remark
All	GTCGCGCCCTCACGGGCGCGTGGATTGAAAC	Canonical sequence
Most	GGCGCGCCCTCACGGGCGCGTGGATTGAAAC	DR1, degenerate variant of terminal repeat
Most	TTCGCGCCCTCATGGGCGCGTGGATTGAAAC	DR2, degenerate variant of the penultimate repeat
FDC 828	TTCGCGCCCTCACGGGCGCGTGGATTGAAAC	Hybrid of DR1 and DR2
LK130-09	TTCGCGCCCTCATGGGCGCGTGGATTGAA_C	
LK169-03	GTCGCGCCCTCACGGGCGCGTGGATTGAAAAC	
LM089-02, LMG 9322, LN003-10, MN10, MN11, MN12	GTCGCGCCCTCACGGGCGCGTGGATTGGAAC	
NCPPB 3610	GTCGCGCCCTCCCGGGCGCGTGGATTGAAAC	

Variant nucleotides in comparison to the canonical DR sequence are double underlined.

## Data Availability

Not applicable.

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
