# Peer review of "Clustered Regularly Interspaced Short Palindromic Repeats in Xanthomonas citri—Witnesses to a Global Expansion of a Bacterial Pathogen over Time"

_microorganisms, 2022, doi:10.3390/microorganisms10091715_

Round 1

Reviewer 1 Report

The manuscript is well-written, I recommend it to accept.

Only two comments:

line10: ...threat..??

line 17: ..based on the...??

Author Response

We are glad that this reviewer enjoyed reading our manuscript and we would like to thank for the time invested in evaluating our manuscript and for catching these two spelling errors, both of which have been corrected:

Line 10: thread -> threat

Line 17: based of -> based on the

Reviewer 2 Report

Dear Authors,

In the work entitled: "Grouped regularly spaced short palindromic repeats in Xanthomonas citri - witnesses of the global expansion of the bacterial pathogen over time", a very interesting, rare topic of research concerning the palondromic distribution of Xanthomonas citri was taken up. This disease has a serious economic impact on citrus production worldwide. The authors generated a spoligotype genealogy, i.e., patterns of absence / presence of CRISPR spacers. As many as 26 new spoligotypes were identified and their probable evolutionary trajectory based on information about the entire genome was constructed. In addition, about 30 additional X. citri patovars were analyzed and it was proved that the oldest part of the CRISPR array was present in the ancestors of several X. citri patovars. The authors of this manuscript provided a framework for further analyzes of the CRISPR loci, which allowed them to draw conclusions about the global spread of the citrus carcinoma pathogen, as exemplified by its two introductions in West Africa. Hence, the work deserves to be published in your journal by all means.

Nevertheless, the authors made a few mistakes that should be corrected before the work appears in print.

Here are the detailed notes:

1.      Lack of a clear goal of the work and no scientific alternative hypothesis to the null hypothesis. The hypothesis should be verified later in the paper.

2.      In the Chapter "Material and methods" there is no sub-chapter "Statistical calculations" where the statistical methods and tests used should be discussed, instead of the chapter "Results".

3.      The chapter "Discussion" is poorly organized. It should be divided into sub-chapters and the results of own research should be better exposed in comparison to the research of other authors.

4.       The application is incomplete. It should contain guidelines for practice and directions for the future.

Author Response

We thank this reviewer for the time invested in evaluating our manuscript and for the detailed notes to help us improve the quality of the manuscript.

  1. Lack of a clear goal of the work and no scientific alternative hypothesis to the null hypothesis. The hypothesis should be verified later in the paper.

We added the specific questions / hypotheses that were addressed in our study at the end of the Introduction (lines 99 – 105), and we come back to these questions in the Discussion (lines 468 – 469, 482, 507 – 507, etc.)

  1. In the Chapter "Material and methods" there is no sub- chapter "Statistical calculations" where the statistical methods and tests used should be discussed, instead of the chapter "Results".

This manuscript does not contain data to be addressed by statistical calculations. Comparison with other typing approaches, such as fingerprinting (AFLP, RAPD, rep-PCR, etc.) or satellite markers, would have been appropriate for statistical calculations. Such an analysis was possible and was performed on a smaller sample for which all data were available from separate previous studies (Jeong et al., 2019). However, such data are not available for this large dataset, and fingerprinting approaches are inherently difficult to compare to each other and cannot be combined on such a large scale.

  1. The chapter "Discussion" is poorly organized. It should be divided into sub-chapters and the results of own research should be better exposed in comparison to the research of other authors.

We introduced sub-chapters to improve comprehensibility of the Discussion.

  1. The application is incomplete. It should contain guidelines for practice and directions for the future.

This manuscript is not meant to provide guidelines for application. However, CRISPR typing will be included in an upcoming revision of the EPPO (European and Mediterranean Plant Protection Organization) Standards on Diagnostics, which will replace the current version PM 7/44(1) (https://www.eppo.int/RESOURCES/eppo_standards/pm7_diagnostics) (https://onlinelibrary.wiley.com/doi/epdf/10.1111/j.1365-2338.2005.00835.x). This revised standard will describe diagnostic protocols for Xanthomonas citri pathovars responsible for citrus bacterial canker disease, including spoligotyping.

Reviewer 3 Report

In this study, Bellanger et al. reports a complete bioinformatic analysis of CRISPR among on X. citri strains. About 516 X. cirti genome sequences were retrieved and reorgnized from Genbank. They were further used to prdict CRISPR loci and to generate the genealogy of the spoligotypes.  In general, they got some novel findings. However, without any molecular experimental data or novel X. citri genome sequence to support the findings, I don't think that it is fit to the requirement of this journal. 

Author Response

We thank this reviewer for the time invested in evaluating our manuscript. Yet, we do not think that addition of another X. citri genome sequence will add any significant information to the manuscript. Indeed, since preparation of the manuscript, one additional genome sequence was released by GenBank: Strain CQ13 from China (BioSample SAMN27753378), accession numbers CP096227 to CP096229. This strain has spoligotype 08, increasing the number of entries with this spoligotype from 120 to 121 (Table S3).

Likewise, performing a molecular experiment, such as PCR amplification and sequencing of the CRISPR locus from a few additional strains, as our laboratory had done in a previous study (Jeong et al., 2019), won’t add any significant information.

According to the MDPI Microorganisms website (https://www.mdpi.com/journal/microorganisms/about), the journal “publishes original scientific research articles, comprehensive reviews, comments, commentaries, perspectives, etc.” from “studies related to prokaryotic and eukaryotic microorganisms, viruses and prions”. The journal’s “aim is to encourage scientists to publish their experimental and theoretical results in as much detail as possible.”

Our study falls into the scope of “Agricultural microbiology, Evolutionary microbiology, Microbial ecology, and Microbial genetics”, as outlined on the MDPI website. Moreover, the Special Issue on “Integrating Science on Xanthomonas and Xylella for Integrated Plant Disease Management” (https://www.mdpi.com/journal/microorganisms/special_issues/xanthomonas_xylella), to which our manuscript should be associated, proposed as potential topics, among others, “diversity, epidemiology, ecology, evolution”. Our manuscript is clearly related to these proposed topics.

Last not least, the same Special Issue has already published another manuscript “without any molecular experimental data or novel … genome sequence” (phrase from the reviewer’s report): Uceda-Campos et al. (2022). Comparative Genomics of Xylella fastidiosa Explores Candidate Host-Specificity Determinants and Expands the Known Repertoire of Mobile Genetic Elements and Immunity Systems. Microorganisms 10: 914. https://doi.org/10.3390/microorganisms10050914

We believe that it is the responsibility of the handling editor to decide whether a manuscript “fits to the requirement of this journal” (phrase from the reviewer’s report).